# The Emerging Role of N-Methyl-D-Aspartate (NMDA) Receptors in the Cardiovascular System: Physiological Implications, Pathological Consequences, and Therapeutic Perspectives

**DOI:** 10.3390/ijms24043914

**Published:** 2023-02-15

**Authors:** Teresa Soda, Valentina Brunetti, Roberto Berra-Romani, Francesco Moccia

**Affiliations:** 1Department of Health Sciences, University of Magna Graecia, 88100 Catanzaro, Italy; 2Laboratory of General Physiology, Department of Biology and Biotechnology “L. Spallanzani”, University of Pavia, 27100 Pavia, Italy; 3Department of Biomedicine, School of Medicine, Benemérita Universidad Autónoma de Puebla, Puebla 72410, Mexico

**Keywords:** NMDA receptors, glutamate, cardiovascular system, cardiomyocytes, vascular smooth muscle cells, endothelial cells, arrhythmia, pulmonary artery hypertension, neurovascular coupling

## Abstract

N-methyl-D-aspartate receptors (NMDARs) are ligand-gated ion channels that are activated by the neurotransmitter glutamate, mediate the slow component of excitatory neurotransmission in the central nervous system (CNS), and induce long-term changes in synaptic plasticity. NMDARs are non-selective cation channels that allow the influx of extracellular Na^+^ and Ca^2+^ and control cellular activity via both membrane depolarization and an increase in intracellular Ca^2+^ concentration. The distribution, structure, and role of neuronal NMDARs have been extensively investigated and it is now known that they also regulate crucial functions in the non-neuronal cellular component of the CNS, i.e., astrocytes and cerebrovascular endothelial cells. In addition, NMDARs are expressed in multiple peripheral organs, including heart and systemic and pulmonary circulations. Herein, we survey the most recent information available regarding the distribution and function of NMDARs within the cardiovascular system. We describe the involvement of NMDARs in the modulation of heart rate and cardiac rhythm, in the regulation of arterial blood pressure, in the regulation of cerebral blood flow, and in the blood–brain barrier (BBB) permeability. In parallel, we describe how enhanced NMDAR activity could promote ventricular arrhythmias, heart failure, pulmonary artery hypertension (PAH), and BBB dysfunction. Targeting NMDARs could represent an unexpected pharmacological strategy to reduce the growing burden of several life-threatening cardiovascular disorders.

## 1. Introduction

Glutamate is the primary excitatory neurotransmitter in the central nervous system (CNS), where it can activate both pre- and post-synaptic ionotropic glutamate receptors (iGluRs) and G-protein coupled metabotropic glutamate receptors (mGluRs) [1]. iGluRs are agonist-operated non-selective cation channels that display different biophysical and pharmacological features and can therefore be subdivided into three structurally distinct classes: α-amino-3-hydroxy-5-methyl-4-isoxasolepropionic acid (AMPA) receptors, kainate receptors, and N-methyl-D-aspartate receptors (NMDARs) [1,2]. While mature AMPA and kainate receptors are only permeable to Na^+^ and K^+^, neuronal NMDARs present high permeability to Ca^2+^ and therefore control many Ca^2+^-dependent functions [1,2]. These include changes in synaptic strength, learning and memory consolidation during cortical development [1,2,3,4], as well as neuronal migration and synaptogenesis during brain development [5]. Sustained activation of NMDARs can result in cytosolic Ca^2+^ overload and exert a neurotoxic effect that can be detrimental to the CNS [6]. Mutations in numerous genes encoding for NMDAR subunits may also result in severe neurological disorders, including intellectual disability, autism spectrum disorders, and schizophrenia [6,7]. A large body of evidence has been produced over the last 15 years to show that NMDARs are expressed and fulfil crucial roles also in non-neuronal cells of the CNS, such as astrocytes [8] and cerebrovascular endothelial cells [9], and in non-neuronal peripheral tissues [10,11], in particular the heart and vasculature [12]. Herein, we focus our attention on the growing role played by NMDARs in the cardiovascular system, in which they regulate heart rate and cardiac contraction, arterial blood pressure, endothelial permeability, and neurovascular coupling (NVC). In parallel, we describe how enhanced NMDAR activity could promote ventricular arrhythmias, heart failure, pulmonary artery hypertension (PAH), and blood–brain barrier (BBB) dysfunction in neurological disorders. Acknowledging the multifaceted role played by NMDARs within the heart and blood vessels can pave the way towards novel therapeutic strategies targeting life-threatening cardiovascular disorders. We do not cover the molecular architecture and role of NMDARs in hematopoietic cells, since these issues have been nicely addressed in two recent review articles [13,14].

## 2. Subunit Composition, Regulatory Mechanisms, and Signaling Modes of Neuronal NMDARs

NMDARs assemble as heterotetrameric non-selective cation channels that can be drawn by multiple subunits: GluN1, GluN2A, GluN2B, GluN2C, GluN2D, GluN3A, and GluN3B [1,15]. Neuronal NMDARs comprise two obligatory GluN1 subunits which combine with either two GluN2 subunits or a mixture of GluN2 and GluN3 subunits [9,10]. Furthermore, GluN1 presents eight isoforms (GluN1-1a–GluN1-4a and GluN1-1b–GluN1-4b) due to alternative splicing, but the functional significance of the differential expression of GluN1 isoforms is still unclear [16]. The subunit composition of NMDARs dictates their biophysical and pharmacological features. Canonical NMDARs consist of two glycine-binding GluN1 subunits and two glutamate-binding GluN2 subunits. GluN1 is also required for correct assembly and trafficking of NMDARs to the plasma membrane [1,2]. Neuronal NMDARs differ from AMPA and kainate receptors not only due to their Ca^2+^ permeability, but also because they undergo a strong inhibition by extracellular Mg^2+^ at negative membrane potentials and because they require the binding of two co-agonists, i.e., glutamate and glycine (or D-serine), for channel activation [1,2,16]. Therefore, NMDARs serve as coincidence detectors that uniquely respond to presynaptic release of glutamate, which opens the channel pore, and postsynaptic depolarization, which relieves Mg^2+^-dependent inhibition [1,2]. At central synapses, the rapid activation of AMPA receptors mediates the inward current that depolarizes the postsynaptic neuron and enables Mg^2+^ unblocking of NMDARs during fast synaptic transmission [1,2]. GluN2A- and GluN2B-containing receptors present high permeability to Ca^2+^ (permeability ratio of Ca^2+^ to Cs^+^ (P_Ca_/P_Cs_) ≈ 7), generate high-conductance single channel openings (50 pS), and are very sensitive to extracellular Mg^2+^ (IC_50_ ≈ 15 µM) [16]. The incorporation of GluN2C or GluN2D dramatically reduces the Ca^2+^ permeability (P_Ca_/P_Cs_ ≈ 4.5), single-channel conductance (37 pS), and sensitivity to Mg^2+^ blockade (IC_50_ ≈ 80 µM) [16]. The recently discovered glycine-binding GluN3 subunit can further change the biophysical properties of NMDARs [17,18]. The GluN3 subunit can replace one of the two GluN2 subunits in classical NMDARs during a narrow time window of postnatal development and in particular cell types, such as pyramidal neurons in the cortex and hippocampus, cerebellar Purkinje neurons, and multiple types of GABAergic interneurons [17,18]. Compared to classical GluN1/GluN2 receptors, tri-heteromeric GluN1/GluN2/GluN3 receptors display lower Ca^2+^ permeability (permeability ratio of Ca^2+^ to the monovalent ions, Na^+^ and K^+^ (P_Ca_/P_M_) of 0.8) and single-channel conductance (27 pS), and are relatively insensitive to Mg^2+^ inhibition at hyperpolarized membrane potentials [18]. Therefore, GluN3A and GluN3B subunits may serve as dominant-negative regulators of NMDARs [17,18]. Intriguingly, tri-heteromeric GluN1/GluN2/GluN3 receptors do not require membrane depolarization to fully open in response to glutamate release [19]. In addition, omission of the glutamate binding GluN2 subunit can result in the assembly of di-heteromeric GluN1/GluN3A or GluN1/GluN3B receptors, which are insensitive to glutamate, extracellular Mg^2+^, and other classical NMDAR inhibitors (e.g., MK-801 or memantine, see below), and function as excitatory glycine-gated channels in heterologous expression systems [1,17]. The expression of functional GluN1/GluN3 receptor complexes in native tissues has been confirmed by Ca^2+^ imaging and electrophysiological recordings of ex vivo mouse optic nerve myelin [20].

The NMDAR complex can be physiologically modulated by multiple endogenous allosteric modulators, including extracellular H^+^ and Zn^2+^, which inhibit NMDARs by targeting GluN1 (Figure 1) and lead (Pb^2+^) cations, which inhibits GluN2A-containing NMDARs [2,11]. Additionally, polyamines, such as spermine and spermidine, exert a voltage-dependent inhibition of NMDARs at high concentrations, whereas they increase the affinity for glycine and promote NMDAR channel opening at micromolar concentrations [11]. A number of synthetic compounds can be exploited to investigate the physiological role of NMDARs with no need for genetic manipulation (Figure 1). These include, of course, the selective agonist NMDA, which recognizes the glutamate-binding site on GluN2, and a number of inhibitors: (1) the competitive antagonist, D-2-amino-5-phosphono-pentanoic acid (AP5), which displaces the agonist from the glutamate binding site; (2) 7-chlorokynurenic acid (7-Cl-Kyna), which is an antagonist for the glycine-binding site on GluN1; (3) MK-801 ((+)-5-methyl-10,11-dihydro-5H-dibenzo[a,d]cyclohepten-5,10-imine maleate), which is an open-channel blocker of NMDAR pores; (4) ifenprodil, which is a selective inhibitor of GluN2B-containing NMDARs; and (5) a variety of substances that directly block NMDARs, including memantine and ketamine [2,11].

Until two decades ago, NMDAR signaling was attributed to its ability to conduct extracellular Ca^2+^ into active spines during fast synaptic transmission and thereby signal coincidence of synaptic inputs in Hebbian plasticity [21,22]. Unexpectedly, numerous independent studies demonstrated that NMDARs may also signal in a flux-independent mode, i.e., without the need to mediate Ca^2+^ influx. It has been demonstrated that glutamate binding to GluN2 may induce conformational changes in the cytoplasmic domain of GluN1 that triggers diverse intracellular signaling pathways, e.g., Ca^2+^ release from the endoplasmic reticulum (ER) and the lysosomal store (Figure 2), protein phosphatase 1 (PP1), and p38 mitogen-activated protein kinase (MAPK) [21,23,24]. Metabotropic NMDAR signaling may be sufficient to induce long-term depression (LTD) and spine shrinkage in the hippocampus [25,26], but has also been reported in rat cortical astrocytes [27,28] and is implicated in synaptic disorders, such as Alzheimer’s disease [29] and schizophrenia [30].

## 3. NMDAR Expression, Signaling, and Function in the Heart

Cardiac excitability and conduction are finely regulated by the autonomic branch of the peripheral nervous system, which innervates the myocardium with both sympathetic and parasympathetic fibers, respectively, releasing the conventional neurotransmitters norepinephrine and acetylcholine [31]. Intriguingly, recent evidence has indicated that another excitatory neurotransmitter system, based upon glutamate and NMDARs, could modulate heart rate, cardiac conduction, and myocardial contraction [32]. Preliminary investigations revealed that GluN1 was expressed in both monkey [33] and rat [34] hearts, in which they were enriched in the conducting system, in atrial and ventricular cardiomyocytes, in the wall of coronary blood vessels, in coronary cardiac intramural nerve terminals, and in cardiac ganglia [32]. Subsequently, GluN1 expression was confirmed in the same anatomic sites of human myocardium [35], which can also express GluN2C [36] and GluN3 [37]. Finally, a recent transcriptomic analysis showed that all GluN subunits are expressed in rat atria [37]. The cardiac distribution of GluN subunits has been summarized in Table 1. These pieces of evidence fostered the view that NMDARs could play a role in the control of cardiac rhythm and excitation.

### 3.1. The Role of NMDAR Activation in Ventricular Arrhythmias and Acute Myocardial Infarction (AMI)

An initial report documented that the systemic (intravenous) infusion of NMDA (125, 250, 500, and 1000 μg/kg) induced a dose-independent increase in rat heart rate that persisted after pharmacological ganglion blockade with hexamethonium [48]. A subsequent investigation evaluated the electrophysiological effects of chronic NMDAR activation [41]. Two weeks of NMDA (3 mg/mL/kg) administration caused a significant increase in heart rate and prolonged the duration of ventricular action potential by downregulating the expression of several K^+^ channels, i.e., K_v_4.2, K_v_4.3, K_v_11.1, and the β-subunit KChIP2, which are involved in cardiac repolarization [41]. In addition, chronic NMDAR activation caused electrical instability, increased the susceptibility to ventricular arrhythmia, and induced myocardial interstitial fibrosis [41]. In accord, K_v_4.2 and K_v_4.3 associate with KChIP2 to mediate the fast transient outward K^+^ current (I_to,fast_), the reduction of which can prolong ventricular action potential duration [49,50]. Similarly, the loss of function of Kv11.1, also known as human ERG, has long been known to result in QT syndrome [50]. However, these pro-arrhythmic effects were reversed by MK-801 [41]. AMI is consequent to the electrophysiological alterations in myocardial substrates caused by myocardial ischemia and ischemia reperfusion, and leads to ventricular fibrillation and tachycardia [12]. D’Amico et al. demonstrated that bocking NMDARs with MK-801, ketamine, or memantine did not affect the incidence of ventricular arrhythmias and mortality induced by coronary occlusion in a rat model of AMI, but significantly reduced the pro-arrhythmic effects of reperfusion [42]. Similar data were obtained by Sun et al., who confirmed that MK-801 mitigated reperfusion-induced ventricular arrhythmias after occlusion of the left anterior descending coronary artery [43]. These findings suggested that cardiac NMDARs could be hyper-stimulated during reperfusion, probably due to excessive release of glutamate observed under these conditions [43,51]. Accordingly, Liu et al. reported that NMDA infusion significantly increased ischemia-reperfusion-induced Ca^2+^ overload in isolated rat ventricular cardiomyocytes, which was translated into a significant loss in ventricular function, cardiomyocytes apoptosis, and myocardial damage in an ex vivo model of AMI [44]. These adverse effects were halted by preventing extracellular Ca^2+^ influx through NMDARs with MK-801 or a Ca^2+^-free buffer [44]. A recent series of reports confirmed that memantine reduced cardiac arrhythmias and infarct size, alleviated cardiac fibrosis and hypertrophy, and partially rescued cardiac contractility in rat models of isoproterenol-induced AMI [52] and heart failure [53]. Finally, NMDARs could be responsible for the high incidence of lethal arrhythmias that occur in healing myocardium owing to sympathetic nerve sprouting (SNS). SNS can increase the cardiac expression of GluN1 in rats, while the subsequent administration of NMDA induced cardiomyocyte apoptosis and triggered ventricular tachycardia and ventricular fibrillation [45]. These effects were again rescued by blocking NMDARs with AP5 [45], thereby confirming that ventricular NMDARs represent a promising molecular target to treat arrhythmias.

### 3.2. The Role of Extracellular Ca^2+^ Entry in NMDA-Induced Ventricular Arrhythmia and Cardiac Remodeling

Multiple mechanisms cooperate to mediate the pro-arrhythmic effects of NMDAR activation in ventricles. An elevation in intracellular Ca^2+^ concentration ([Ca^2+^]_i_) is likely to play a major role in NMDAR-dependent myocardial pathogenesis [41,54]. Glutamate can increase the [Ca^2+^]_i_ in rat cardiomyocytes by activating NMDARs in the presence of extracellular Mg^2+^ and in the absence of the co-agonist D-serine (Figure 3A) [46,55]. Therefore, NMDAR activation is able to depolarize ventricular (as well as atrial, see Section 3.3) cardiomyocytes both directly (i.e., by conducting Na^+^ and Ca^2+^ into the cytosol) and indirectly (i.e., through the Na^+^/Ca^2+^ exchanger (NCX), which can be activated by a subplasmalemmal Ca^2+^ rise) [56,57]. Thus, a NMDAR-dependent increase in [Ca^2+^]_i_ could increase the risk of trigger activity and arrhythmias. It is still unclear whether extracellular Ca^2+^ entry through NMDARs can stimulate Ca^2+^-induced Ca^2+^ release (CICR) through type 2 ryanodine receptors (RyR2) (Figure 3B) or increase the sarcoplasmic reticulum (SR) Ca^2+^ load in a Sarco-Endoplasmic Reticulum Ca^2+^-ATPase (SERCA) 2a (SERCA2a)-dependent manner, as previously shown for lysosomal two-pore channels [58,59]. In neonatal rat ventricular cardiomyocytes, GluN2B has been detected in the Z-disc, in which it colocalizes with RyR2 on the juxtaposed SR cisternae (Figure 3A) [40]. Moreover, both GluN1 and GluN2B were identified in the outer mitochondrial membrane and inner mitochondrial cristae of rat hearts (Figure 3A) [39]. It is therefore reasonable to speculate that glutamate stimulates RyR2-dependent Ca^2+^ release and mitochondrial Ca^2+^ overload (Figure 3B), which would lead to mitochondrial depolarization and opening of the mitochondrial permeability transition pore (mPTP) [60].

In addition, NMDAR-mediated intracellular Ca^2+^ signals could modulate gene expression and thereby drive the down-regulation of K_v_4.2, K_v_4.3, K_v_11.1, and KChIP2 [41], as well as SERCA2a [43] and proteins (Figure 3B). In agreement with this hypothesis, activation of the Ca^2+^-dependent transcription factor NF-κB [61,62] can reduce I_to,fast_ by down-regulating the expression of K_v_4.2, K_v_4.3, K_v_11.1, and KChIP2 in neonatal rat ventricular cardiomyocytes [63]. The reduction in SERCA2a levels in hypertrophic and failing hearts could also involve Ca^2+^-dependent regulatory pathways [64]. Finally, NMDA-induced myocardial fibrosis, which may predispose ventricles to arrhythmias [57], can also by triggered by intracellular Ca^2+^ overload. In accord, many reports have demonstrated that NMDAR stimulation can induce mitochondrial depolarization, reactive oxygen species (ROS) production, and caspase-3-mediated apoptosis in rat ventricular cardiomyocytes (Figure 3A) [46,47]. Similarly, NMDARs promoted Ca^2+^ influx and Ca^2+^-dependent apoptosis by enhancing p38 MAPK activity in primary human neonatal cardiomyocytes cultured under ischemic conditions [65]. In line with these findings, infusion of MK-801 at the beginning of the reperfusion strongly attenuated oxidative stress in a rat model of AMI [66], whereas co-administration of glutamate and glycine dampened the cardioprotective effects of verapamil by increasing ROS production [67]. Intriguingly, a recent report showed that the anthracycline anticancer drug, pirarubicin, induced Ca^2+^ overload, oxidative stress, and apoptosis through GluN2D-containing NMDARs in HL-1 (mouse) and H9C2 (rat) cardiomyocytes [68].

### 3.3. The Role of NMDARs in Atrial Excitability and Conductivity and Their Involvement in Atrial Fibrillation

A recent series of studies have focused on the effect of NMDAR activation in the atria, where they are quite abundant, particularly in the sino-atrial node [35]. In addition, glutamate concentration was found to be enhanced in the left atrial appendages of patients [69]. In agreement with these observations, Xie et al. demonstrated that an intrinsic glutamatergic system is present in rat atrial cardiomyocytes [37]. They found that glutamate-containing vesicles are abundantly located beneath the plasma membrane. Moreover, these cells express glutaminase, which generates glutamate from glutamine; the excitatory amino acid transporter, EAAT1; and many iGluR isoforms, including GluN1 [37]. Accordingly, both glutamate and NMDA elicit transient inward currents and reduce the action potential threshold in rat atrial cardiomyocytes [37]. Furthermore, MK-801 and AP5 attenuate the atrial conduction velocity both ex vivo and in vivo [37]. Finally, an intrinsic glutamatergic system was also detected in human atria, and NMDARs were found to mediate transient inward currents and control also the conduction velocity in cultured human induced pluripotent stem cell (hiPSC)-derived atrial cardiomyocyte monolayers [37]. 

The evidence described above led to the hypothesis that exaggerated NMDAR activation could also contribute to abnormal excitability and conductivity in atrial fibrillation. In accord, a transcriptomic analysis conducted on a canine model of atrial fibrillation revealed that several genes encoding for glutamate signaling components were targeted by the differentially expressed microRNAs (miRNAs) of the DLK1-DIO3 locus [70]. Furthermore, Shi et al. has previously shown that intraperitoneal injection of NMDA (3 mg/kg) in rats reduced heart rate variability, which is associated with higher adverse outcome of AMI [71], by causing a dramatic sympathovagal imbalance, with a depressed vagal tone and an enhanced sympathetic activity in rats [38]. In addition, NMDA administration caused atrial electrical remodeling and atrial fibrillation by inducing atrial fibrosis and down-regulating connexin 40 through an increase in metalloproteinase 9 (MMP-9) expression [38]. Notably, Xie et al. confirmed that blocking NMDARs with MK-801 and AP5 prevented the occurrence and halted the progression of atrial fibrillation in vivo [37]. Thus, atrial NMDARs could provide a novel molecular target to design novel therapeutic strategies to treat atrial fibrillation, which is the most widespread persistent cardiac arrhythmia [32].

### 3.4. The Endogenous Agonist of Cardiac NMDARs: Is There A Role for Homocysteine (Hcy)?

Given the evidence that NMDARs are expressed in the heart and control cardiac excitation and conduction, the question as to their physiological agonist is obviously raised. A functional glutamatergic signaling system has been uncovered in atria (see Section 3.3), but not in ventricles. No evidence has been provided in favor of glutamate release from sympathetic nerves, as initially proposed by some authors [46]. It has been suggested that the peripheral organs, including the heart and peripheral vasculature, are exposed to baseline extracellular levels [10], which fluctuate between 5 and 100 µM/L in the plasma and 150 and 300 µM/L in the whole blood [72]. As anticipated above, glutamate concentrations ranging between 10 and 200 µM/L elicit robust Ca^2+^ signals in cardiomyocytes [55], thereby suggesting that cardiac NMDARs could be stimulated by circulating glutamate. However, it has long been known that astrocytes release glutamate via multiple mechanisms, including exocytosis, connexin 43 hemichannels, P2X Purinoreceptor 7 receptors, and bestrophin-1, to regulate neuronal activity [73]. Future work might explore the possibility that ventricular cardiomyocytes release glutamate not only during myocardial ischemia, but also during evoked activity or in response to β-adrenergic stimulation. 

Alternately, cardiac NMDARs could be sensitive to the endogenous levels of Hcy, a derivative of the hepatic L-methionine metabolism that may serve as an NMDAR agonist by associating with the glutamate binding sites when glycine concentrations are raised beyond 50 µM [54,74]. Intriguingly, hyperhomocysteinemia (HHcy), which results from excessive serum levels of Hcy, can lead to coronary artery disease, AMI, ventricular arrhythmias, sudden cardiac death, and atherosclerosis [54,75]. The following pieces of evidence support the emerging notion that circulating Hcy could target cardiac NMDARs. First, a pharmacological blockade of NMDARs with MK-801 [76] or a genetic deletion of cardiac GluN1 [77] reduced the mitochondrial production of ROS and reactive nitrogen species, thereby impairing MMP9 activation and mPTP opening and halting the decline in myocyte contractility. Second, a genetic deletion of cardiac GluN1 also inhibited mitophagy [78], which can become extremely harmful to cardiac tissue when it becomes chronic [79]. Third, the combined application of Hcy, or Hcy thiolactone, NMDA, and glycine, caused a decrease in mechanical cardiac function and an increase in oxidative stress that were rescued by intracoronary infusion of memantine [80,81]. Quinolinic acid, a downstream metabolite of tryptophan that activates NMDARs in the CNS, has also been proposed as endogenous agonist of peripheral NMDARs [10], but its effect on cardiac NMDARs is still unknown. Thus, while the endogenous agonist (i.e., glutamate) engaging atrial NMDARs is coming of age, the signaling molecules that physiologically gate ventricular NMDARs still await to be fully elucidated. 

## 4. The Role of NMDARs in VSMCs and Vascular Endothelial Cells

The main role of peripheral vasculature is to carry oxygen and nutrients to distant organs and tissues, from which carbon dioxide and catabolic waste are then removed and disposed through the urinary, respiratory, or gastrointestinal systems [82]. Large conduit arteries deliver oxygenated blood to the branched capillary networks that effect the exchanges of respiratory gaseous and nutrients/catabolites between circulating blood and the surrounding tissues. Resistance arteries (lumen diameter ∼150–250 μm) and arterioles (<150 μm) convey the blood from large conduit arteries to downstream capillaries and represent the major site of vascular resistance [82]. The vascular wall is mainly composed of endothelial cells, which line the inner lumen of blood vessels and are directly exposed to blood stream, and a surrounding layer of VSMCs. Capillaries comprise only a monolayer of endothelial cells, while VSMC contraction in resistance arteries and arterioles is a crucial determinant of blood pressure [82,83]. However, endothelial cells release multiple vasoactive mediators that are able to control VSMC contractility and thereby induce vasorelaxation (e.g., NO, prostaglandins, and hydrogen sulfide) or vasoconstriction (e.g., endothelin) [82,83]. A preliminary report showed that GluN1 was present both in rat aorta, the largest artery in the body, and in rat mesenteric resistance arteries [48]. Then, GluN1 was detected in rat carotid arteries and GluN1, GluN2A, GluN2B, GluN2C, and GluN2D were identified in rat aortic endothelial cells (RAECs) (Table 2) [84] and in rat aortic smooth muscle (A7r5) cells (Table 2) [85]. GluN1, GluN2A, GluN2B, GluN2C, and GluN2D were also found in human pulmonary arterial smooth muscle cells (hPASMCs) (Table 2) and human pulmonary microvascular endothelial cells (hPMVECs) in combination with the postsynaptic density protein-95 (PSD-95) (Table 2) [86]. In agreement with these observations, NMDA induced AP5-sensitive inward currents in the presence of glycine and in the absence of extracellular Mg^2+^ in hPASMCs, thereby confirming that NMDARs were functional [87]. Multiple GluN subunit- and NMDAR-mediated intracellular Ca^2+^ signals were also recorded in mouse [9,88,89] and human [24,90] cerebrovascular endothelial cells, as more widely discussed in Section 5.

### 4.1. The Hemodynamic Effect of NMDAR Activation in Peripheral Vasculature

The role of peripheral NMDARs in the regulation of blood pressure is yet fully unraveled [12]. Systemic infusion of NMDA (125, 250, 500, and 1000 μg/kg) induced a dose-dependent increase in rat mean arterial pressure (MAP) that was not affected by hexamethonium (see also Section 3.1) and was therefore mediated by vascular NMDARs [48]. In accord, the hemodynamic response to NMDA was sensitive to AP5 and HA-966, which are antagonists of the glycine binding site on GluN1. It was then found that NMDARs activate neuronal, rather than endothelial, NO synthase (nNOS) to generate NO, and that NO release drives NMDA-dependent vasoconstriction [48]. These unexpected findings suggested that peripheral NMDARs also engage NADPH oxidase (NOX), as described in neurons [91], so that nNOS-derived NO reacts with NOX-derived ROS to form peroxynitrite and induce vasoconstriction [12]. A follow-up study showed that ethanol attenuates an NMDAR-mediated increase in ROS production and MAP [92]. A recent series of studies demonstrated that nNOS is also expressed and contributes to NO signaling in VSMCs and endothelial cells and that it is quite abundant in the mesenteric bed [93]. Intriguingly, NMDARs recruit the endothelial NO synthase (eNOS) and mediate vasodilation in mouse brain microcirculation [94,95], and NMDAR-dependent vasorelaxation has also been reported in rat coronary circulation [66,80]. These findings suggest that NMDARs could be coupled to different NOS isoforms in different vascular districts, e.g., nNOS in aorta and mesenteric resistance arteries and eNOS in brain and coronary microvessels. The hypothesis that peripheral NMDAR signaling could be exaggerated in essential hypertension, as suggested for PAH (see below), cannot be discounted [12]. Future work will have to assess whether dysregulated NMDARs are located in VSMCs, in vascular endothelial cells, or both.

### 4.2. The Role of Peripheral NMDARs in Vascular Remodelling during HHcy-Induced Atherosclerosis and in PAH

Vascular cells are exposed to circulating glutamate, which could directly target endothelial cells and may easily reach VSMCs. In addition, vascular cells can be directly stimulated by plasma Hcy and HHcy can induce endothelial dysfunction and accelerate myointimal hyperplasia and luminal narrowing, thereby resulting in atherosclerosis [54,75,96]. A preliminary study showed that HHcy increased NMDAR expression and stimulated proliferation in A7r5 cells [85]. These effects were mediated by MMP-9 and interleukin-1β, which support VSMC proliferation and migration, and rescued by MK-801 [85]. A subsequent study confirmed that HHcy stimulated human VSMCs to release MMP-2, which is involved in remodeling and destabilization of the atherosclerotic plaques by recruiting the phosphatidylinositol 3-kinase (PI3K) and MAPK phosphorylation cascades [97]. These pro-atherosclerotic effects were again alleviated by MK-801 and L-glycine, which favors NMDAR internalization [97]. It was further demonstrated that HHcy stimulates the expression of C-reactive protein (CRP), which also promotes VSMC proliferation and migration by inducing ROS production, MAPK activation, and the nuclear translocation of NF-κB in rat aortic VSMCs [98]. The primary role of NMDARs in HHcy-dependent CRP expression and secretion was underpinned by the inhibitory action of MK-801 [98]. Future work will have to assess whether endothelial NMDARs are involved in HHcy-induced endothelial dysfunction, as suggested in [12]. Nevertheless, peripheral NMDARs stand out as novel molecular targets to interfere with HHcy-induced vascular remodeling and atherosclerotic progression.

On the other hand, a recent study has undoubtedly unveiled the pathogenic role of vascular NMDARs in PAH [86]. Early reports had already shown that tissue-type plasminogen activator (tPA) [99] and urokinase plasminogen activator (uPA) [100] increase lung arterial contractility and promote lung vascular permeability by activating NMDARs in mice. These studies showed that preventing the interaction between pulmonary NMDARs and tPA or between uPA and a docking peptide that binds either of these enzymes and interferes with their occupancy of the NMDAR binding site can mitigate pulmonary injury [99,100]. More recently, Dumas et al. documented that glutamate levels were enhanced in pulmonary arteries isolated from PAH patients as compared to control arteries, and that glutamate accumulation was more robust in medial lesions [86], which are featured by VSMC hyperplasia. The increase in glutamate production was secondary to the accumulation of glutamine, which is the glutamate precursor in the glutaminase reaction, and was more evident in pulmonary arterioles (<150 µm) [86], which represent the main target of current therapies [101]. This study also showed that GluN1 protein was up-regulated and hyperphosphorylated at Ser-896 in PAH pulmonary arteries, particularly in VSMC lesions. This latter feature was rather relevant since phosphorylation of Ser-896 increases the surface expression of GluN1 in neurons [102]. Furthermore, many components of the neuronal glutamatergic system were identified in hPASMCs and hPAMECs, including glutaminase 1 (GLS1) and GLS2, which catalyze glutamate production from glutamine [103]; the vesicular glutamate transporter, VGLUT1; and the scaffolding protein, PSD-95. Interestingly, GLS1 and VGLUT1 were enhanced in PAH arteries and both hPASMCs and hPMVECs were able to release glutamate in a Ca^2+^-dependent manner. Additionally, both endothelin-1 and platelet-derived growth factor-BB (PDGF-BB), which accelerate vascular remodeling in PAH, stimulated GluN1 phosphorylation and induced hPASMC proliferation. GluN1 phosphorylation was spatially confined within the regions of cell-to-cell contact between hPASMCs [86]. Therefore, the authors speculated about a quasi-synaptic mode of glutamatergic communication via NMDARs between adjacent hPASMCs in both culture and PAH pulmonary arteries (Figure 4). In line with these findings, hypoxia failed to promote pulmonary vascular remodeling and pulmonary hypertension in transgenic mice selectively lacking GluN1 in VSMCs. Moreover, blocking NMDARs with MK-801 in a rat model of monocrotaline-induced PAH alleviated remodeling of pulmonary arteries and the corresponding decrease in right ventricle contractility [86]. A follow-up study showed that PDGF-BB could increase the surface expression of GluN2B in hPASMCs via Src-dependent phosphorylation, thereby resulting in the assembly of antiproliferative and antimigratory NMDARs [104]. However, GluN2B expression was down-regulated in PAH pulmonary arteries, which further highlights the crucial role played by VSMC NMDARs in medial hyperplasia and vascular obstruction [104]. Therefore, this study could pave the way for novel therapeutic avenues to treat PAH by targeting pulmonary NMDARs [101]. A crucial issue that remains to be solved concerns the requirement for complete Mg^2+^ unblocking to measure NMDAR-mediated inward currents in hPASMCs [87]. NMDAR activation exacerbates phenylephrine-induced contractile responsiveness of human pulmonary arteries under hypoxic, but not normoxic, conditions [87]. However, the membrane potential of hypoxic pulmonary arterial SMCs is more depolarized as compared to normoxic cells due to the reduced expression and activity of voltage-gated K^+^ channels [105] and to the activation of the acid-sensing ion channel 1 [106]. The persistent depolarization of hypoxic pulmonary arterial SMCs could be involved in the glutamate-dependent activation of NMDARs and support vascular remodeling and vasoconstriction in PAH.

## 5. The Role of NMDARs in Cerebro-Microvascular Endothelial Cells (CMVECs)

Glutamate is the major excitatory neurotransmitter in the CNS and modulates neuronal excitability and synaptic plasticity via iGluRs and mGluRs. It is therefore not surprising that AMPA receptors and NMDARs, as well as the G_q/11_-protein coupled mGluR1 and mGluR5, are widely expressed also in the other main components of the neurovascular unit, i.e., astrocytes and microvascular endothelial cells [8,9,10,107,108,109,110,111]. Cerebral microvessels, i.e., arterioles and capillaries, receive an extensive glutamatergic innervation [112,113] and CMVECs can respond to neuronal activity via NMDARs [94,114,115]. In accord, NMDAR subunits have been detected in human [24,90,115] and rodent [95,114,115] CMVECs, in which GluN1 is preferentially located to the basolateral membrane surface to directly perceive synaptically released glutamate [116]. As illustrated in Table 3, CMVECs express GluN1, GluN2A, GluN2C, and GluN3A in mouse cortical arteries [94,116], whereas GluN1, GluN2B, and GluN3A were detected in spinal cord microvascular endothelial cells [90]. The molecular architecture of endothelial NMDARs in human CNS microvessels has been investigated by exploiting primary CMVECs, which express GluN1 [115,117], and in the hCMEC/D3 cell line, which has been established from the human temporal lobe microvessels [118] and express GluN1, GluN2C, and GluN3B subunits (Table 3) [24]. Therefore, the molecular configuration of endothelial NMDARs in cortical vessels is enriched with GluN2C and GluN3 subunits, which are likely to yield a low sensitivity to extracellular Mg^2+^, low single-channel conductance, and sensitivity to glycine [10,24,119].

NMDARs regulate multiple functions of the BBB [10]. The BBB is mainly formed by CMVECs that line brain microvessels and is therefore instrumental to maintain CNS homeostasis and ensure proper neuronal signaling. In addition, the BBB protects brain parenchyma from blood-borne pathogens and toxins and its restrictive nature represents a major obstacle for drug delivery to the CNS [120,121]. Hogan-Cann and Anderson have recently surveyed the involvement of endothelial NMDARs in the control of the BBB in health and disease [10]. In this section, we focus our attention on the latest investigations describing the role of endothelial NMDARs in the fine regulation of BBB permeability and NVC, i.e., the mechanism whereby an increase in neuronal activity is translated into an increase in cerebral blood flow (CBF) [112,122], and their implication in neurological disorders.

### 5.1. Endothelial NMDARs Control BBB Permeability

The movement of solutes across the BBB between circulating blood and brain parenchyma can occur through a paracellular or transcellular route [120,121]. Glutamate can induce the opening of the BBB and thereby increase paracellular permeability via NMDAR-mediated intracellular Ca^2+^ signals [120,125,126,127,128]. In accord, NMDA-induced elevation in BBB permeability was sensitive to MK-801 [125,126,127] or AP5 [126,128] and an NMDA-evoked increase in [Ca^2+^]_i_ was recorded in both human [24,115,126] and mouse [129,130] CMVECs and in mouse cerebral artery endothelial cells [88]. Intriguingly, NMDA-induced intracellular Ca^2+^ signals in hCMEC/D3 cells are sensitive to AP5 and genetic knockdown of GluN1, but not to MK-801 and to removal of extracellular Ca^2+^ [24]. In addition, whole-cell patch-clamp recordings showed that NMDA did not elicit any detectable inward current both in the absence and in the presence of extracellular Mg^2+^ [24]. Nevertheless, this investigation revealed that NMDARs may operate in a flux-independent mode to signal the increase in [Ca^2+^]_i_ in hCMEC/D3 cells. In accord, NMDARs interact with mGluR1 and mGluR5 to mobilize lysosomal Ca^2+^ via TPCs and ER-stored Ca^2+^ via InsP_3_ receptors and to induce store-operated Ca^2+^ entry (Figure 2) [24,131]. Conversely, NMDA-induced intracellular Ca^2+^ signals in mouse CMVECs disappear in the absence of extracellular Ca^2+^ and in the presence of the pore-blocking antagonist, MK-801 [88]. Therefore, NMDARs can signal the increase in [Ca^2+^]_i_ in a flux-dependent manner in mouse CMVECs and in a flux-independent (i.e., metabotropic manner) in human CMVECs. NMDAR-mediated intracellular Ca^2+^ signals can increase the BBB permeability by inducing NO release [24,108,128] and myosin light chain (MLC) phosphorylation through the Ca^2+^-dependent recruitment of Rho-kinase (ROCK) and MLC kinase (MLCK) [108]. Finally, extracellular Ca^2+^ entry through NMDARs also controls the transcellular route across the mouse BBB by recruiting Ca^2+^/calmodulin-dependent protein kinase II (CaMKII), protein kinase C (PKC), and RhoA [115].

### 5.2. Endothelial NMDARs Control NVC

The mechanism whereby an increase in neuronal activity may lead to a local elevation in CBF to match the high energy demand of active neurons with a supply of oxygen and nutrient supply from the blood is known as NVC or functional hyperemia [112,113]. According to the canonical model, NVC is triggered by the release of vasoactive mediators, such as NO, arachidonic, potassium ions (K^+^), from neurons and astrocytes. These signals target the contractile cells, i.e., VSMCs and pericytes, of brain microvessels to induce vasorelaxation and increase CBF [112,113]. A series of recent studies, reviewed in [9,10,89,132,133,134], demonstrated that CMVECs may sense neuronal activity and thereby elicit a robust increase in CBF through a blend of ion channels that are located in the plasma membrane and include the inward rectifier K^+^ (K_ir_2.1) channel [135], Transient Receptor Potential Ankyrin 1 (TRPA1) channels [136], and NMDARs [10]. In accord, the Anderson group demonstrated that the stimulation of endothelial NMDARs with glutamate induced arteriolar relaxation in ex vivo mouse middle cerebral arteries through the Ca^2+^-dependent release of NO [94,95,116]. Astrocytes contribute to the activation of basolateral endothelial NMDARs by releasing the co-agonist D-serine in response to an increase in [Ca^2+^]_i_ [95,116]. As expected (see Section 5), glutamate-evoked arteriolar vasodilation was not sensitive to Mg^2+^-dependent blockades [94]. In agreement with these observations, the hemodynamic response to sensory stimulation was largely compromised in a transgenic mouse model selectively lacking endothelial NMDARs [114] and eNOS [137]. In addition, the full hemodynamic response to somatosensory stimulation in mouse arteries and arterioles requires the stimulation of luminal NMDARs by circulating tPA [138]. Intriguingly, in the human cerebrovascular endothelial cell line hCMEC/D3, NMDARs may signal in a flux-dependent manner to induce NO release, do not require the presence of D-serine, and are not sensitive to Mg^2+^-dependent inhibition [24]. 

The evidence that endothelial NMDARs at the NVU are stimulated during glutamatergic transmission is also relevant for synaptic events that go beyond the hemodynamic response. Early work has shown that the hippocampal long-term potentiation (LTP) is inhibited in eNOS-deficient mice [139,140]. Intriguingly, eNOS is not expressed in CA1 pyramidal cells [141], which clearly indicates the crucial role of endothelium-derived NO in synaptic plasticity and memory formation. Future experiments should assess whether LTP is impaired in transgenic mice selectively lacking endothelial NMDARs and eNOS not only in the hippocampus, but also in other brain areas, such as the cortex, amygdala, and cerebellum. The role of vascular NMDARs in endothelial-to-neuronal communication has been suggested by a recent study showing that the vessel-associated migration of GABA interneurons along the pial migratory route is finely regulated by endothelial NMDARs [124].

### 5.3. The Pathogenic Role of Endothelial NMDARs in the CNS 

Disruption of the BBB is a hallmark of many neurological disorders that are associated to hypersynchronized neuronal activation and/or excessive glutamate release, such as traumatic brain injuries (TBI), epilepsy, multiple sclerosis, and stroke [120,121,142]. Furthermore, the BBB permeability is significantly increased in vascular dementia and other neurodegenerative disorders such as Alzheimer’s disease (AD) due to the enhanced oxidative stress imposed to cerebrovascular endothelial cells [143]. Early studies conducted on cultured CMVECs showed that endothelial NMDARs could contribute to the disruption of BBB integrity through a variety of mechanisms, including the altered expression of BBB efflux transporters (e.g., ABCG2) [117,144], the decreased expression and relocalization of tight junctions protein (e.g., occludin and claudin-5) [108,111], MMP-2, and MMP-9 release [129]. In addition, excessive doses of glutamate (1 mM) induced endothelial dysfunction through NMDAR-mediated production of ROS [125,127,145] and peroxynitrite [146], thereby disrupting the BBB integrity in vitro. In accord, an increase in endothelial [Ca^2+^]_i_ not only engages the eNOS, but it can also stimulate ROS generation through the Ca^2+^-dependent NOX4 [147], which is widely expressed in CMVECs [130]. 

A recent series of investigations confirmed that excessive stimulation of endothelial NMDARs can increase the BBB permeability in several animal models of neurological disorders. Aberrant glutamate release during epileptic seizures increased vascular permeability in the rat cerebral cortex via endothelial NMDARs (Table 4) [128]. Similarly, endothelial-specific deletion of the regulator of G-protein signaling 5 (RGS5) enhanced endothelial NMDAR signaling in a mouse model of brain stroke, which exacerbated the increase in the BBB permeability and the severity of the ischemic damage (Table 4) [108]. The serin-protease tPA has been shown to disrupt the BBB integrity and favor monocyte infiltration in mouse models of multiple sclerosis (Table 4) [148]. Blocking NMDARs with a monoclonal antibody (known as Glunomab) that selectively targets GluN1 inhibited tPA-induced leukocyte transmigration in vitro and halted the progression of neurological impairments in a mouse model of multiple sclerosis (i.e., autoimmune encephalomyelitis) in wild-type but not in tPA-deficient animals [149]. Moreover, NMDARs were found to mediate tPA-dependent monocyte transmigration through the human BBB via extracellular signal-regulated kinase 1 activation [123]. In addition, the endovascular infusion of Glunomab reduced the incidence of intracellular aneurysms by blocking tPA-dependent activation of endothelial NMDARs in mice (Table 4) [150]. These preliminary findings strongly support the notion that targeting endothelial NMDARs could represent a promising therapeutic strategy to attenuate BBB dysfunction in neurological disorders as severe as epilepsy, multiple sclerosis, and stroke [151]. In addition, intraluminal infusion of amyloid-β_(1–40)_ reduced extracellular Ca^2+^ entry through endothelial NMDARs and attenuated endothelium-dependent dilation in mouse cerebral arteries (Table 4) [88]. Thus, defective endothelial NMDAR signaling could contribute to the impaired hemodynamic response (and possibly to loss in neuronal regulation by endothelium-derived NO) in neurodegenerative diseases such as AD. Finally, anti-NMDAR autoimmune encephalitis is a rare neurological disorder that is caused by autoreactive anti-NMDAR autoantibodies targeting GluN1 and produced by B lymphocytes [151,152]. This in turn results in NMDAR hypo-responsiveness and leads to a decline in cognitive function and neuropsychiatric symptoms [153]. Future work should assess whether anti-NMDAR autoantibodies also target the endothelial NMDARs, as suggested by the defect in BBB integrity that has been reported in both animal models [152] and patients [154] affected by this disease.

## 6. Conclusions

A burgeoning body of evidence has demonstrated that NMDARs do not only fulfil a crucial function in the CNS, but also in the peripheral organ systems. The regulatory role of NMDARs in the cardiovascular system is also coming of age. In the heart, NMDARs are located within the conduction system and in the working myocardium. Therefore, they can regulate both the cardiac rhythm and cardiac excitability in a healthy heart, while exaggerated NMDAR signaling could contribute to atrial fibrillation, ventricular arrhythmias, and heart failure. In the peripheral vasculature, NMDARs are expressed in VSMCs and endothelial cells, but it is still unclear whether and how they regulate MAP. Preliminary evidence suggests that the regulation of local blood flow could be differentially affected by peripheral NMDARs, depending on the vascular bed and on their location in VSMCs vs. vascular endothelial cells. In pulmonary circulation, VSMC NMDARs are certainly involved in the pathogenesis of PAH, although their role in hPMVECs is still unclear. The role of endothelial NMDARs has been clearly unveiled in cerebral microcirculation, where they regulate NVC and BBB permeability. Conversely, it is still unknown whether they are expressed in cerebral VSMCs and pericytes. As various neurological disorders depend on NMDAR subunit mutations [6,7], the possibility that also endothelial NMDARs are involved in such genetic diseases cannot be discounted. Importantly, pre-clinical studies demonstrated that the pharmacological blockade of NMDARs could represent an effective strategy in treating life-threatening cardiovascular pathologies, ranging from arrhythmic disorders to PAH and neurological/neurodegenerative diseases. The molecular composition of NMDARs in cardiomyocytes (Table 1), VSMCs and vascular endothelial cells (Table 2), and in CMVECs (Table 3) is yet to be fully elucidated. The complement of ion channels endowed to VSMCs and endothelial cells may extensively vary depending on the vascular district [156,157,158,159]. Thus, it is likely that distinct GluN subunits will be found to contribute to vessel-associated NMDARs in different vascular beds. Understanding the molecular make-up of NMDARs is mandatory to decipher the diverse array of biophysical features they present in the multiple cellular components of the cardiovascular system (e.g., Mg^2+^-dependent inhibition in hPASMCs but not in ventricular cardiomyocytes and CMVECs) and to fully exploit their therapeutic potential.

## Figures and Tables

**Figure 1 ijms-24-03914-f001:**
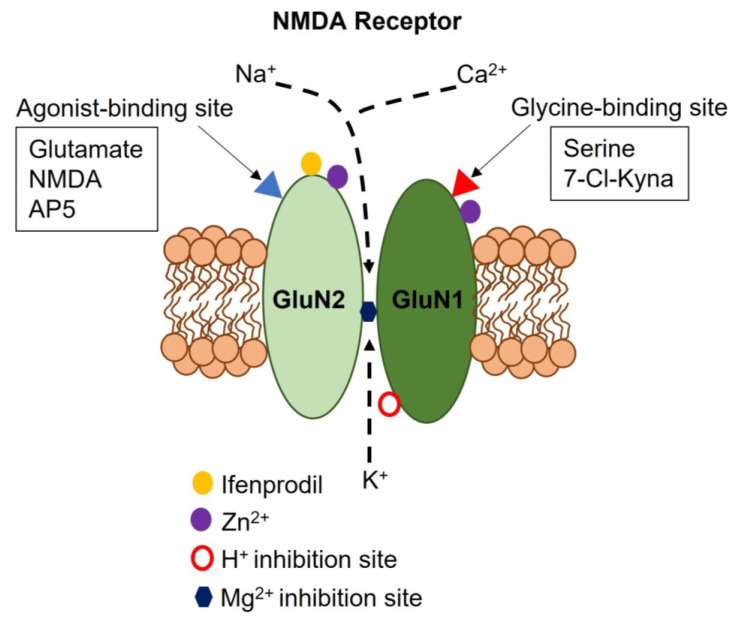
Structure of the neuronal NMDAR. Neuronal NMDARs are heterotetramers comprising two obligatory GluN1 subunits and two GluN2 subunits. The GluN1 subunit contains the agonist-binding site, whereas the GluN2 subunit contains the glycine (or serine) binding site. NMDARs are permeable to intracellular K^+^ (outward direction) and extracellular Na^+^ and Ca^2+^ (inward direction). The NMDAR channel pore contains a Mg^2+^ binding site that must be relieved by AMPA receptor-mediated membrane depolarization to enable glutamate-induced NMDAR activation. Extracellular Zn^2+^ binds to both GluN1 and GluN2, whereas ifenprodil selectively binds to GluN2B subunits to enhance H^+^-mediated NMDAR inhibition. The incorporation of GluN3 subunits (not shown) confers slightly different biophysical features to the NMDAR, including a low Mg^2+^-dependent inhibition, a reduced Ca^2+^ permeability, and a lower single-channel conductance.

**Figure 2 ijms-24-03914-f002:**
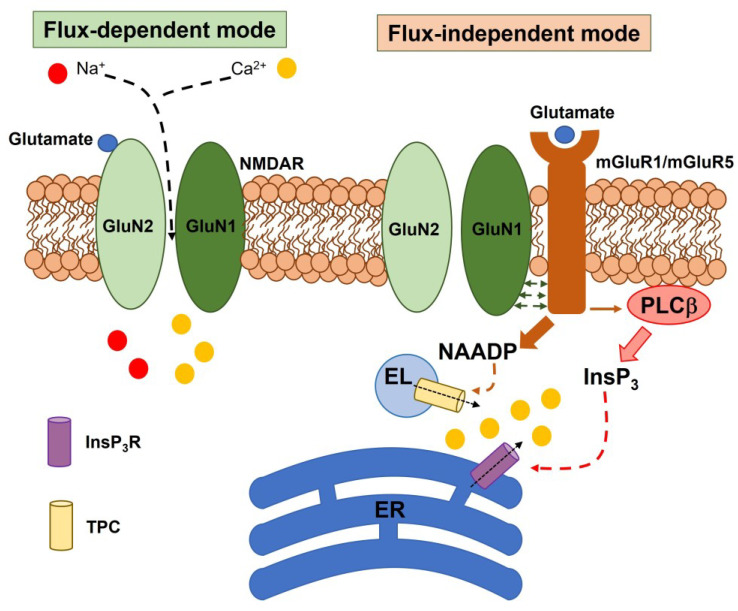
NMDARs may signal in a flux-independent mode to elicit intracellular Ca^2+^ release. NMDARs can signal in either a flux-dependent or independent mode [24,27]. Ionotropic NMDARs mediate the influx of extracellular Na^+^ and Ca^2+^ and induce membrane depolarization or recruit Ca^2+^-dependent decoders such as nNOS (not shown). Metabotropic NMDARs can interact with mGluR1 and mGluR5 to stimulate PLCβ and promote InsP_3_-induced ER Ca^2+^ release. In addition, metabotropic NMDAR signaling involves the NAADP-induced mobilization of lysosomal Ca^2+^. The reduction in ER Ca^2+^ concentration may in turn lead to the activation of SOCE on the plasma membrane, not shown.

**Figure 3 ijms-24-03914-f003:**
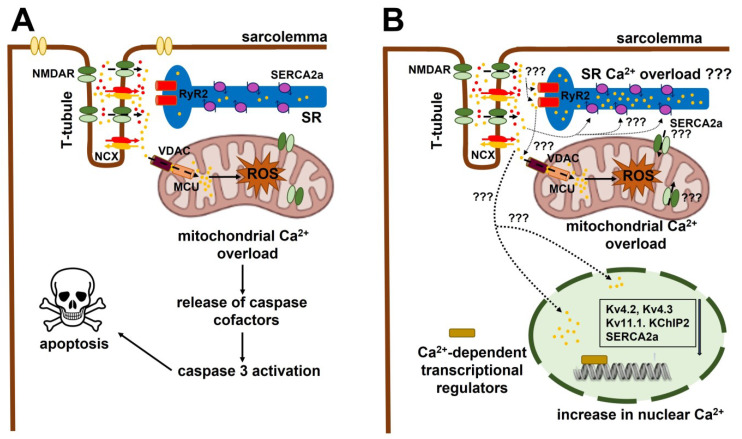
NMDAR-mediated extracellular Ca^2+^ entry may result in mitochondrial Ca^2+^ overload and caspase 3-dependent apoptosis in ventricular cardiomyocytes. (**A**) Extracellular Ca^2+^ entry through NMDARs may result in mitochondrial Ca^2+^ overload by flowing through voltage-dependent anion channels (VDAC) and mitochondrial Ca^2+^ uniporters (MCU), which are located in the outer and inner mitochondrial membranes, respectively. The mitochondrial Ca^2+^ overload may, in turn, lead to the formation of the mitochondria permeability transition pore (mPTP) (not shown), release of caspase cofactors, caspase 3 activation, and cell death. In addition, NMDAR opening can induce depolarization via the influx of Na^+^ and Ca^2+^, which could promote further depolarization via activation of the Na^+^/Ca^2+^ exchanger (NCX). Recent evidence detected NMDARs in the Z-disc, which is juxtaposed to the T-tubules. (**B**) Extracellular Ca^2+^ entry through NMDARs in ventricular cardiomyocytes could either promote CICR via RyR2 (denoted by “???”) or lead to SR Ca^2+^ overload in a SERCA2a-dependent manner (denoted by ???). We hypothesize that SR-dependent Ca^2+^ release via RyR2 could further contribute to mitochondrial Ca^2+^ overload (denoted by “???”). Finally, NMDAR-mediated extracellular Ca^2+^ entry could also induce an increase in nucleoplasmic Ca^2+^ and thereby down-regulate the expression of K_v_4.2, K_v_4.3, K_v_11.1, and KChIP2 through Ca^2+^-dependent transcriptional regulators (denoted by “???”).

**Figure 4 ijms-24-03914-f004:**
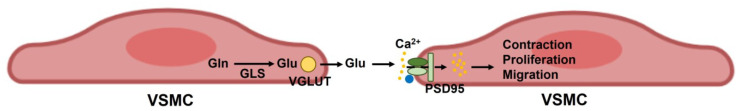
Model of NMDAR-mediated quasi-synaptic communication between adjacent VSMCs in pulmonary circulation. A recent investigation by Dumas and coworkers [86] proposed a quasi-synaptic mode of glutamatergic communication via NMDARs between adjacent VSMCs in human pulmonary circulation. According to this mode, glutamate (Glu) is formed directly from glutamine (Gln) by glutaminase (GLS1 and/or GLS1) and is thereafter released in the extracellular space through the vesicular glutamate transporter (VGLUT). Herein, Glu binds to juxtaposed NMDARs that are located on adjacent VSMCs and stimulates the influx of extracellular Ca^2+^, which in turn promotes VSMC contraction, proliferation, and migration. It has been suggested that the PDZ scaffolding protein, PSD95, contributes to recruit, anchor and cluster NMDARs on the plasma membrane.

**Table 1 ijms-24-03914-t001:** NMDARs in the heart.

Chamber	Species	Subunit Expression	Proposed NMDAR Composition	Role	References
**Atrium**					
	Human	GluN1, GluN3	Not defined	Excitability and conductivity	[35,37]
	Rat	GluN1, GluN2A, GluN2B, GluN2C, GluN2D, GluN3A, GluN3B	Not defined	Excitability and conductivity, fibrillation	[34,37,38]
	Monkey	GluN1	Not defined	Not investigated	[33]
**Conduction system**					
	Human	GluN1	Not defined	Not investigated	[35]
	Rat	GluN1	Not defined	Not investigated	[34]
	Monkey	GluN1	Not defined	Not investigated	[33]
**Ventricle**					
	Human	GluN1	Not defined		[35]
	Rat	GluN1, GluN2B	Not defined	Delayed repolarization, arrhythmias, fibrosis, apoptosis, mitochondrial dysfunction	[34,39,40,41,42,43,44,45,46,47]
	Monkey	GluN1	Not defined	Not investigated	[33]

Lin et al. reported the expression of GluN2C transcript in whole cardiac lysates from human samples [36].

**Table 2 ijms-24-03914-t002:** NMDARs in the vascular system.

Cell Type	Species	Subunit Expression	Proposed NMDAR Composition	Role	References
Aortic VSMCs	Rat	GluN1, GluN2A, GluN2B, GluN2C, GluN2D	Not defined	Proliferation, vasoconstriction, MMP-9 and IL-1β expression, reduction in basal NO release	[48,85]
hPASMCs	Human	GluN1, GluN2A, GluN2B, GluN2C, GluN2D	Not defined	Proliferation, hypoxia-induced pulmonary hypertension and vascular remodeling	[86,87]
Aortic endothelial cells	Rat	GluN1, GluN2A, GluN2B, GluN2C, GluN2D	Not defined	Proliferation	[85]

Abbreviations: hPASMCs: human pulmonary arterial smooth muscle cells; IL-1β: interleukin-1β; MMP-9: metalloproteinase 9; NO: nitric oxide; VSMCs: vascular smooth muscle cells.

**Table 3 ijms-24-03914-t003:** NMDARs in cerebrovascular endothelial cells.

Cell Type	Species	Subunit Expression	Proposed NMDAR Composition	Role	References
Endothelial cell	Human	GluN1, GluN2C, GluN3A	Not defined	NO release, increase in transcellular and paracellular permeability	[24,90,115]
Endothelial cell	Mouse	GluN1, GluN2A, GluN2B, GluN3A	Not defined	NO release, increase in transcellular and paracellular permeability, monocyte transmigration, GABA interneuron migration	[90,94,95,114,115,116,123,124]

**Table 4 ijms-24-03914-t004:** The involvement of endothelial NMDARs in cerebrovascular disorders.

Disease	Altered Function	Pharmacological Targeting	References
Epilepsy	BBB disruption	AP5	[128]
Stroke	BBB disruption	MK-801	[108]
Intracerebral aneurysm	BBB disruption	Glunomab	[150]
Multiple sclerosis	BBB disruption and monocyte infiltration	Glunomab	[148,149]
HHyc	BBB disruption	Memantine	[155]
AD	Reduced arterial vasodilation	Not assessed	[88]

Abbreviations: AD: Alzheimer’s disease; BBB: blood–brain barrier; HHyc: hyperhomocysteinemia.

## Data Availability

Not applicable.

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
