# Peer review of "The Emerging Role of N-Methyl-D-Aspartate (NMDA) Receptors in the Cardiovascular System: Physiological Implications, Pathological Consequences, and Therapeutic Perspectives"

_ijms, 2023, doi:10.3390/ijms24043914_

Round 1

Reviewer 1 Report

The authors should be praised for this review, as it is very extensive and well-constructed. I started reading and could not stop until I finished. Actually, I learned quite a lot by reading this manuscript. Frankly, I hope it is published soon as I would love to use the  figure in the review on the flux-independent mode in teaching physiology. This concept is not yet presented in general physiology textbooks.

Two topics came to mind while reading through the manuscript that the authors may wish to address.

1.    Line 190: “However, these pro-arrhythmic effects were reversed by MK-801 [41].”
MK-801---- is this not a broad Ca2+ channel blocker and can even block pre-synaptic voltage-gated Ca2+ channels?

2.    The widely read book and even a NETFLIX movie “Brain on Fire” is about the autoimmune disorder where one develops that antibody to the NMDA receptor. I am not sure if it is to a particular subunit or to a complex. It could be interesting to note if or suggest that data be examined if these individuals also suffer from cardiovascular issues due to the possibility of the NMDA in the vascular also being targeted.

Reviewer 2 Report

1) glutamate-gated ionotropic receptors. This is not true. The correct definition is ligand-gated ion channels that are activated by the neurotransmitter glutamate.

2) A figure showing the role of calcium ions and mitochondrial overload with NMDAR activation should be presented. Those. figure 2 should at least reflect the role of calcium in pathologies. I would recommend making a separate chapter in the article devoted to these effects.

Round 2

Reviewer 2 Report

The article can be accepted for publication in its current form